# Harnessing Dendritic Cell Function in Hepatocellular Carcinoma: Advances in Immunotherapy and Therapeutic Strategies

**DOI:** 10.3390/vaccines13050496

**Published:** 2025-05-04

**Authors:** Shiding Ying, Haiyan Liu, Yongliang Zhang, Yu Mei

**Affiliations:** 1Department of Systems Biology, School of Life Sciences, Southern University of Science and Technology, Shenzhen 518055, China; 12112325@mail.sustech.edu.cn; 2Department of Microbiology and Immunology, Yong Loo Lin School of Medicine, National University of Singapore, Singapore 117545, Singapore; micliuh@nus.edu.sg; 3NUSMED Immunology Translational Research Programme, Yong Loo Lin School of Medicine, National University of Singapore, Singapore 117456, Singapore; 4Immunology Programme, Life Science Institute, National University of Singapore, Singapore 117456, Singapore

**Keywords:** hepatocellular carcinoma, dendritic cells, immunotherapy, tumor microenvironment, antigen presentation, antitumor immunity

## Abstract

Hepatocellular carcinoma (HCC) is a major cause of cancer-related mortality worldwide. Conventional therapies are frequently limited by tumor heterogeneity and the immunosuppressive tumor microenvironment (TME). Dendritic cells (DCs), central to orchestrating antitumor immunity, have become key targets for HCC immunotherapy. This review examines the biological functions of DC subsets (cDC1, cDC2, pDC, and moDC) and their roles in initiating and modulating immune responses against HCC. We detail the mechanisms underlying DC impairment within the TME, including suppression by regulatory T cells (Tregs), myeloid-derived suppressor cells (MDSCs), tumor-associated macrophages (TAMs), and cancer-associated fibroblasts (CAFs). Additionally, we discuss novel DC-based therapeutic strategies, such as DC-based vaccines designed to enhance antigen presentation and T cell activation. Combining DC vaccines with immune checkpoint inhibitors (ICIs), including PD-1/PD-L1 and CTLA-4 blockers, demonstrates synergistic effects that can overcome immune evasion and improve clinical outcomes. Despite progress, challenges related to DC subset heterogeneity, TME complexity, and patient variability require the further optimization and personalization of DC-based therapies. Future research should focus on refining these strategies, leveraging advanced technologies like genomic profiling and artificial intelligence, to maximize therapeutic efficacy and revolutionize HCC treatment. By restoring DC function and reprogramming the TME, DC-based immunotherapy holds immense potential to transform the management of HCC and improve patient survival.

## 1. Introduction

Hepatocellular carcinoma (HCC) is one of the leading causes of cancer-related mortality worldwide, highlighting the critical need for more effective therapeutic strategies. Despite advances, the efficacy of conventional treatments, including surgery, liver transplantation, and systemic therapies, is often limited by the aggressive nature of HCC, its cellular heterogeneity, and the highly immunosuppressive tumor microenvironment (TME). Recent breakthroughs, particularly the establishment of immune checkpoint inhibitor (ICI) combinations, such as atezolizumab plus bevacizumab and durvalumab plus tremelimumab as standard first-line treatments, have significantly improved survival outcomes for patients with advanced HCC [1]. Furthermore, novel strategies combining these systemic agents with locoregional therapies, such as transcatheter arterial chemoembolization (TACE), are emerging as effective approaches for intermediate-stage disease [2]. Despite these advances, significant challenges persist: many patients do not respond to current ICI-based regimens, treatment resistance frequently develops, and the immunosuppressive TME continues to hinder treatment efficacy. Addressing the limitations of current therapies motivates the exploration of alternative approaches; harnessing the unique capabilities of dendritic cells (DCs) represents a promising immunotherapeutic direction. DCs are critical mediators for initiating and modulating antitumor immunity, rendering them attractive targets for immunotherapy. These cells play an indispensable role in capturing, processing, and presenting antigens to T cells, thereby orchestrating adaptive immune responses. Furthermore, the unique ability of DCs to modulate the balance between immune activation and tolerance is crucial for overcoming the challenges posed by the TME. Modulating DC functions holds potential for enhancing effective antitumor immune responses. Therapeutic strategies aiming to improve DC activation, enhance antigen presentation, and reprogram the TME may provide new avenues for overcoming the challenges in HCC treatment. Despite this potential, significant barriers hinder the effective application of DC-based therapies in HCC, including the complexity of DC subsets, the intricate nature of the TME, and the need for more precise therapeutic strategies. Future research must focus on optimizing these approaches, integrating them with other immunotherapeutic modalities, and tailoring treatments to individual needs to maximize clinical benefits.

## 2. Biological Functions of Dendritic Cells and Their Role in Cancer Immunity

### 2.1. Subsets of DCs

Dendritic cells are specialized antigen-presenting cells (APCs) that act as a bridge between innate and adaptive immunity. These cells are categorized into distinct subpopulations based on phenotypic markers, developmental pathways, and functional properties. Each subpopulation exhibits significant differences in infiltration and function within the TME, influencing tumor progression through multiple mechanisms (Figure 1) [3,4].

cDC1: The cDC1 subset plays a critical role in antitumor immunity. Its differentiation is regulated by transcription factors, including interferon regulatory factor 8 (IRF8) and inhibitor of DNA binding 2 (ID2) [5]. Signature markers for cDC1 include BDCA3 (CD141) in humans and CD103 and CD8α in mice [6]. cDC1 efficiently activates CD8^+^ cytotoxic T lymphocytes (CTLs) via the cross-presentation of antigens derived from apoptotic or necrotic tumor cells [7]. Studies have shown that cDC1-deficient mice display impaired immune responses to tumors and significantly reduced therapeutic efficacy of T cell-mediated tumor immunotherapies, including immune checkpoint blockade (ICB) therapy [8,9]. A high cDC1 density in tumor tissues correlates with favorable prognoses across multiple solid tumors, including HCC, melanoma, non-small cell lung cancer (NSCLC), and ovarian cancer, emphasizing its pivotal role in anti-tumor immunity [10].

cDC2: The cDC2 subset of DCs plays a crucial yet complex role in shaping the anti-tumor immune response. While traditionally recognized for their ability to prime CD4^+^ T helper cells that support cytotoxic T cell responses against cancer, recent research highlights a dual capacity of cDC2s: they can also contribute to the induction and maintenance of regulatory T cells (Tregs), thereby potentially suppressing anti-tumor immunity. Understanding this delicate balance is critical for developing effective cancer immunotherapies.

cDC2s are highly efficient at capturing and presenting exogenous antigens, including those derived from tumors, primarily to CD4^+^ T cells via MHC class II molecules [11]. Upon activation in a pro-inflammatory environment, typically characterized by the presence of danger signals and specific cytokines, cDC2s can effectively prime naive CD4^+^ T cells to differentiate into various effector T helper subsets, such as Th1 and Th17 cells [12]. Th1 cells are particularly important in anti-tumor immunity as they produce cytokines like interferon-gamma (IFN-γ) that enhance the activity of cytotoxic CD8^+^ T cells, the primary killers of cancer cells. Th17 cells can also contribute to anti-tumor responses through the recruitment of other immune cells. This pathway represents the immunogenic facet of cDC2 function, promoting a robust anticancer immune response. Studies have shown that cDC2s are vital for priming anti-tumor CD4^+^ T cells in various tumor models [3].

However, the TME is often enriched with immunosuppressive factors, including certain cytokines (e.g., IL-10 and TGF-β) and metabolic conditions, which can significantly influence DC function [13,14]. In this altered state, cDC2s can adopt a more tolerogenic phenotype. Instead of driving effector T cell differentiation, they can promote the induction of Tregs. In the context of cancer, an increased presence and activity of Tregs within the TME can suppress the function of effector T cells, including tumor-specific CTLs, thereby allowing the tumor to evade immune surveillance and progress. cDC2s can contribute to Treg induction through various mechanisms, including the presentation of specific antigens in a non-inflammatory context, the production of immunosuppressive cytokines, and the expression of co-inhibitory molecules. The plasticity of cDC2s in response to tumor-released environmental cues has been observed, leading to the acquisition of immunosuppressive phenotypes [15].

This dual capacity of cDC2s underscores their critical position at the intersection of immunity and tolerance in the context of cancer. Their ultimate effect on the anti-tumor response depends heavily on the signals they receive from their microenvironment. A pro-inflammatory milieu favors their role in promoting effector T cells, while an immunosuppressive TME can skew them towards inducing Tregs.

Melanoma is often characterized by substantial immune infiltration and is commonly referred to as an ‘inflamed’ or ‘hot’ tumor. In this context, the TME is rich in pro-inflammatory cytokines like IFN-γ and TNF-α, secreted by infiltrating immune cells such as NK cells and activated T cells. Intratumoral cDC2s within this environment efficiently capture tumor-derived antigens. The robust inflammatory signals drive their maturation, leading to the upregulation of co-stimulatory molecules and MHC class II. These activated cDC2s then migrate to the regional lymph nodes to induce the differentiation of naive CD4^+^ T cells into effector Th1 cells. These tumor-specific Th1 cells, armed with the ability to produce IFN-γ and IL-2, subsequently traffic back to the tumor, where they significantly enhance the activity of tumor-infiltrating cytotoxic CD8^+^ T cells, contributing to tumor cell lysis and control [16].

Conversely, within immunosuppressive TMEs, such as those frequently observed in cancers like pancreatic carcinoma, the function of cDC2s can be significantly skewed towards promoting tolerance. These environments are characterized by high concentrations of immunosuppressive factors, including TGF-β, IL-10, and vascular endothelial growth factor (VEGF), often accompanied by a dense stromal barrier and metabolic constraints. In such settings, cDC2s exhibit impaired maturation, reduced expression of co-stimulatory molecules, and increased expression of inhibitory ligands such as programmed death ligand 1 (PD-L1) [11]. The cytokine profile of the immunosuppressive TME favors the induction and expansion of Tregs.

This inherent plasticity and dual capacity of cDC2s highlight their central position in the complex interplay between the tumor and the immune system. Consequently, a comprehensive understanding of the specific molecular and cellular cues within the TME that dictate cDC2 polarization towards either immunogenic or tolerogenic functions is therefore paramount. Manipulating this balance, either by directly targeting cDC2 function to enhance effector priming or by remodeling the immunosuppressive TME to favor cDC2 maturation and immunogenicity, represents a promising avenue for improving the efficacy of current and future cancer immunotherapies.

pDC: Plasmacytoid dendritic cells (pDCs) are known for their potent type I interferon (IFN-I) secretion capacity [17]. Their differentiation is driven by the transcription factors IRF8, RUNX1, and TCF4 [18]. In humans, this subset is identified by the expression of BDCA2 (CD303), CD123, and BDCA4 (CD304) [19]. These markers help distinguish pDCs from other DC subsets, highlighting their unique functional role in immune regulation. pDCs exhibit a dual role in TME: they can promote antitumor immunity by inducing cDC1 maturation and enhancing the effector functions of CD8^+^ T cells and NK cells, while simultaneously supporting tumor progression through immunosuppressive molecules (e.g., PD-L1, ICOSL, and IDO) [20]. In HCC, pDCs often exhibit a dysfunctional phenotype characterized by impaired production of IFN-Is and increased expression of immunosuppressive molecules, such as PD-L1 and ICOSL [21]. This dysfunction contributes to immune evasion by promoting Treg expansion and suppressing cytotoxic T cell responses. Studies have shown that impaired pDC functions correlate with poor prognosis in HCC patients [22], underscoring the importance of strategies to restore or enhance pDC’s anti-tumor activity in HCC as a potential therapeutic approach.

moDC: Monocyte-derived dendritic cells (moDCs) arise under inflammatory conditions and exhibit characteristics overlapping with monocytes and cDC2. Their differentiation is dependent on IRF4 [23]. Recent studies have highlighted the ability of moDCs to activate CD8^+^ T cells through cross-presentation and their significant role in enhancing the efficacy of anti-PD-1 checkpoint inhibitors, particularly in combination therapies [24]. These combination strategies often include cytotoxic T-lymphocyte-associated protein 4 (CTLA-4) blockade, chemotherapy, radiotherapy, or cancer vaccines, which collectively enhance antigen presentation and T cell activation, thereby improving antitumor responses.

### 2.2. Activation of Dendritic Cells in Hepatocellular Carcinoma

The activation of DCs is a pivotal step in initiating anti-tumor immune responses during the development of HCC. DCs recognize pathogen-associated molecular patterns (PAMPs) or damage-associated molecular patterns (DAMPs) through surface pattern recognition receptors (PRRs), which subsequently activate signaling pathways mediated by adaptor proteins such as TRIF and MyD88, leading to phenotypic and functional changes in DCs [25].

#### 2.2.1. Pattern Recognition and Signaling Pathways

PAMPs are conserved components of pathogenic microorganisms, such as lipopolysaccharide (LPS) and viral nucleic acids, whereas DAMPs are endogenous molecules released from damaged or dead cells, including heat shock proteins (HSPs), high mobility group protein 1 (HMGB1), etc. Toll-like receptors (TLRs) are capable of sensing various PAMPs and DAMPs, triggering downstream signaling pathways [26,27]. For instance, TLR4 recognizes LPS, a well-established classical DC stimulator, subsequently initiating DC activation [28]. In HCC, HMGB1 released by tumor cells after chemoradiotherapy can be recognized by DC through TLR4, which induces DC maturation and anti-tumor immune responses [29]. In addition, tumor-derived DNA sensed by DCs can induce IFN-β secretion via the cGAS-STING pathway, a mechanism further confirmed in murine HCC models [30,31]. Recent advancements in multifunctional nanomaterial research have highlighted the potential of metal–organic frameworks (MOFs), such as MOF-801, to act as STING agonists. These materials have been shown to activate the cGAS-STING/NF-κB signaling pathway through TLR4, thereby promoting DC maturation and inducing the secretion of IL-6, thus inhibiting tumor growth [32].

#### 2.2.2. Migration and Antigen Delivery

DCs undergo a tightly regulated process of migration and antigen delivery that bridges peripheral immune sensing with adaptive immune activation. Immature DCs in non-lymphoid tissues efficiently capture antigens and upregulate chemokine receptor CCR7, which enables them to migrate to lymphoid organs, such as lymph nodes [33]. During migration, DCs undergo progressive maturation, significantly enhancing their antigen-presenting capacity. In lymph nodes, mature DCs present antigenic peptides to naive T cells via MHC molecules, delivering the primary signal. Concurrently, co-stimulatory molecules such as CD80 and CD86 on DCs bind to CD28 on T cells, providing the secondary activation signal [34]. To complete the activation process, mature DCs secrete cytokines, including IL-12, that drive the differentiation of T cells into effector T cells, delivering the third and final signal necessary for robust T cell activation [35,36].

#### 2.2.3. Immunomodulatory Role in HCC

DC activation plays an important role in immunomodulation of HCC, regulating both pro-tumor immunity and anti-tumor immunity. Activated DCs inhibit tumor progression through antigen presentation to promote Th1 cells differentiation, and recruit tumor-specific lymphocytes into the TME [37]. Additionally, DCs secrete IL-12, which activates NK cells, further amplifying anti-tumor immune responses [35]. Meanwhile, under the influence of the TME, DCs can also inhibit the function of effector T cells by activating Tregs and myeloid-derived suppressor cells (MDSCs), which in turn leads to tumor immune escape.

The functional status of DCs in HCC immunotherapy is often assessed by evaluating the expression of activation markers such as CD40, CD80, CD83, CD86, CCR7, and MHC [38]. These markers reflect the maturation state of DCs and their ability to mediate immune responses effectively. Collectively, assessing the activation status of DCs in HCC is fundamental to understanding their role in antitumor immunity and provides guidance for optimizing DC-based immunotherapies.

## 3. Impaired DC Functions in the TME

### 3.1. Inhibitory Effect of TME Components on DC

In HCC, the TME exerts profound immunosuppressive effects on DCs, significantly impairing their antigen-presenting functions and antitumor efficacy. Several components of the TME contribute to these inhibitory effects, including Tregs, MDSCs, tumor-associated macrophages (TAMs), and cancer-associated fibroblasts (CAFs).

Tregs: Tregs play a pivotal role in suppressing DC activity within the TME. One major mechanism involves CTLA-4, which is highly expressed on Tregs. CTLA-4 binds to CD80/CD86 on the surface of DCs with a higher affinity than that of CD28 [39], blocking CD28-mediated costimulation and impairing DCs’ ability to activate effector T cells. Additionally, CTLA-4-signaling downregulates CD80/CD86 expression on DC surface [40,41], indirectly upregulates PD-L1 expression [34], further limiting DC antigen-presenting capacities.

Tregs also modulate DC metabolism by inducing the production of indoleamine 2,3-dioxygenase (IDO), which converts tryptophan into pro-apoptotic metabolites, thereby dampening effector T cell activation [42]. Lymphocyte activation gene-3 (LAG-3), another inhibitory molecule expressed on Tregs, interacts with MHC class II molecules on DCs, diminishing their ability to stimulate T cells [43]. Moreover, Treg-expressed CD27 binds to CD70 on effector T cells, interfering with CD70-mediated costimulatory signaling [44]. Furthermore, Treg-derived exosomes, enriched with immunosuppressive cytokines such as IL-10 and TGF-β, can reprogram DCs into a tolerogenic phenotype. This shift reduces DC activation of effector T cells while promoting Treg expansion [45]. Studies have shown that the inhibition of Treg infiltration or the blocking of their key molecular products (e.g., CTLA-4 or IL-10) significantly enhances DC-mediated anti-tumor immune responses [46,47], confirming the suppressive effect of Tregs on DCs’ antigen-presenting capacity in TME.

MDSCs: MDSCs are actively recruited into the TME through the action of various chemokines, including CCL2, CCL5, CCL26, CXCL8, and CXCL12, as well as other mediators such as granulocyte macrophage colony-stimulating factor (GM-CSF), IL-6, or prostaglandin E2 (PGE2) [14]. Upon recruitment, MDSCs can interfere with DC functionality by transferring oxidized lipids, which subsequently accumulate within DCs as lipid vesicles. These vesicles form covalent bonds with HSP70, disrupting the transmembrane translocation of antigenic peptide-major histocompatibility complex I (pMHC-I) complexes, thereby impairing DC-mediated antigen cross-presentation [48,49]. Higher levels of MDSCs in HCC tumor-bearing mice are associated with increased IL-10 expression, reduced IL-12 production by DCs, and lower T cell stimulatory activity [50]. Together, these mechanisms compromise the antigen-processing efficiency of DCs, thereby facilitating cancer immune evasion within the TME.

TAMs: TAMs within the TME can inhibit the antigen-presenting capacity of DCs through multiple direct and indirect mechanisms. Influenced by cytokines such as PGE2, TAMs preferentially adopt an anti-inflammatory phenotype by secreting immunosuppressive cytokines, including IL-10 and TGF-β, which dampen the maturation process of DCs, thus impairing their antigen capture, processing, and presentation efficiency [51]. Specifically, IL-10 inhibits DCs from secreting IL-12, thereby weakening their capacity to activate effector T cells, particularly CD8^+^ T cells [52]. Additionally, TAMs can attenuate DC-T cell interactions by orchestrating the spatial distribution of key chemokines within the TME, thereby facilitating the infiltration of immunosuppressive cell populations. Specifically, TAM-derived CCL22 recruits Tregs, which suppress effector T cell activity, while the secretion of CCL2 and CXCL8 promotes the accumulation of MDSCs, further dampening antitumor immune responses. This chemokine-mediated immunosuppressive network disrupts effective antigen presentation and T cell priming, ultimately compromising antitumor immunity [51]. In addition, studies have shown that after depletion of TAMs by macrophage colony-stimulating factor receptor (CSF1R) inhibitors in mice, the IL-12 secretory capacity of DCs and their effector T cell stimulatory function were restored [53]. These findings suggest that TAMs inhibit DC function in TME through multiple mechanisms, thereby driving tumor immune escape.

CAFs: CAFs in the TME could impair DC migration and antigen-processing capabilities, thereby promoting tumor immune evasion. CAFs can secrete factors such as Wnt Family Member 2 (WNT2) and vascular endothelial growth factor (VEGF), which inhibit the migration of DCs to lymph nodes or T cell-enriched regions, thereby hindering effective antigen presentation and activation of adaptive immunity [54,55]. Furthermore, CAFs can upregulate tryptophan-2,3-dioxygenase (TDO2), a critical enzyme in tryptophan metabolism, to compromise the antigen-processing ability of DCs and induce a tolerogenic phenotype [56].

### 3.2. DC Dysfunction in Virus-Associated HCC

Chronic hepatitis B virus (HBV) infection, a major etiological factor for HCC, is characterized by profound immune dysregulation, including DC dysfunction. Accumulating evidence indicates that both moDCs and pDCs from individuals with HBV exhibit impaired phenotypic and functional maturation. These defects include the reduced expression of costimulatory molecules (CD80, CD86, and HLA-DR) and the diminished secretion of IL-12, which collectively compromise T cell priming and cytotoxic effector responses [57]. Concurrently, HBV-associated DCs produce higher levels of immunosuppressive mediators such as nitric oxide and IL-10, further contributing to immune tolerance and viral persistence [58].

HBV surface antigen (HBsAg) has been shown to interact directly with DCs, leading to aberrant activation. While it can promote partial maturation through sCD14 and TLR4-dependent pathways, the resulting DCs exhibit a tolerogenic phenotype with impaired IL-12 production and weakened T cell stimulation [59]. Moreover, transcriptomic profiling has revealed a downregulation of IL6ST (gp130), a core component of IL-6 family cytokine signaling, in DCs from chronic HBV patients. Experimental restoration of IL6ST in vitro partially reversed the functional defects of these DCs, suggesting a mechanistic link between HBV infection and disrupted cytokine receptor signaling [60]. These alterations in DC function not only facilitate immune evasion by HBV but also create an immunologically permissive environment that may accelerate hepatocarcinogenesis.

### 3.3. Broader Immunosuppressive Mechanisms in the TME

In HCC, the TME serves as a complex and dynamic ecosystem that fosters the immune escape ability of the tumor. This is mediated by a combination of immunosuppressive cells, co-inhibitory signaling pathways, soluble factors, and metabolic alterations [61].

#### 3.3.1. Immunosuppressive Signaling Pathways in the HCC TME

The TME of HCC is characterized by a multifaceted immunosuppressive network that facilitates tumor immune evasion. While Tregs and MDSCs contribute to immune suppression through cytokine secretion, their effects are significantly amplified by inhibitory signaling pathways and soluble mediators [62]. Co-inhibitory pathways, particularly the PD-1/PD-L1 and CTLA-4 axes, play a pivotal role in dampening antitumor immunity by inducing T cell exhaustion and suppressing effector T cell activation [63,64]. Notably, PD-1/PD-L1 interactions impair cytotoxic T cell function, while CTLA-4 competes with CD28 for costimulatory signals, further limiting immune activation. Beyond cell-intrinsic signaling, the HCC TME is enriched with immunosuppressive soluble factors, including IL-10, TGF-β, GM-CSF, VEGF, and MCP-1, which orchestrate the recruitment, differentiation, and functional reprogramming of immunosuppressive cell populations [65]. These factors activate oncogenic signaling cascades such as STAT3 and NF-κB, which not only suppress effector T cell responses but also promote stromal remodeling, angiogenesis, and metastatic progression. Collectively, these mechanisms establish a profoundly immunosuppressive TME that facilitates tumor progression and resistance to immunotherapeutic interventions.

#### 3.3.2. Tumor-Derived Exosomes

One significant contributor to the immunosuppressive TME is the presence of tumor-derived exosomes. These small vesicles secreted by tumor cells carry a variety of immunosuppressive factors that can alter DC function and hinder their ability to effectively activate T cells. Tumor-derived exosomes deliver molecules such as PD-L1, TGF-β, miRNAs, and lipid metabolites that significantly impact DC behavior [66]. For example, PD-L1 on tumor-derived exosomes binds to PD-1 on DCs, leading to a suppression of DC maturation and function, which ultimately dampens T cell activation. Similarly, TGF-β delivered from exosomes promotes the polarization of DCs into an immunosuppressive phenotype, reducing their capacity to activate effector T cells. Furthermore, miRNAs such as miR-21 and miR-155, carried by exosomes, have been shown to regulate the differentiation and function of DCs, favoring the expansion of immunosuppressive cell populations like MDSCs and limiting T cell responses [67]. Additionally, lipid metabolites within tumor-derived exosomes, such as PGE2, can influence DC maturation and function, contributing to the suppression of antitumor immunity [68]. Collectively, these tumor-derived exosomes effectively reprogram DCs in the TME, impairing their antigen-presenting ability and promoting an environment conducive to tumor immune evasion. By interfering with DC functions, exosomes not only prevent effective immune responses but also contribute to the overall immunosuppressive landscape in HCC [69].

#### 3.3.3. Metabolic Dysregulation

Metabolic alterations in the TME play a crucial role in suppressing immune responses in HCC. These alterations in metabolites significantly affect the activity of immune cells. Due to the high metabolic rate of tumor cells, the TME is often characterized by a low-oxygen, high-lactate environment that directly inhibits effector T cells and NK cells, further enhancing immune escape. Hypoxia induces the expression of hypoxia-inducible factor (HIF), promoting the recruitment and activation of Tregs and MDSCs. Simultaneously, lactic acid accumulation suppresses the proliferation and cytokine secretion of effector T cells [70,71]. Nutrient depletion, such as glucose and glutamine competition, further impairs T cell metabolism in the TME, diminishing their activity and facilitating tumor immune escape under immune escape [72,73].

The immunosuppressive TME in HCC arises from a complex interplay of cellular, molecular, and metabolic factors, each contributing to immune evasion through distinct yet interconnected mechanisms. Notably, DCs, as central players in immune regulation, are profoundly affected by these suppressive signals. A comprehensive understanding of the molecular basis underlying these mechanisms is essential for designing novel immunotherapeutic strategies aimed at restoring DC functionality and enhancing antitumor immunity in HCC.

DCs, pivotal orchestrators of anti-tumor immunity, are also profoundly affected by these metabolic constraints. Within the nutrient-depleted and hypoxic TME, DCs undergo metabolic reprogramming that skews them towards a tolerogenic phenotype. Elevated lipid uptake, mediated by receptors like CD204, disrupts MHC class I-mediated antigen presentation and impairs T cell priming [74]. Furthermore, a shift in DCs from glycolysis to fatty acid oxidation (FAO)—driven by PPARs and β-catenin signaling—facilitates Treg induction and inhibits cytotoxic T cell responses [75]. Metabolites such as lactate and adenosine reinforce this suppressive state by downregulating pro-inflammatory cytokines like IL-12 and upregulating immunoregulatory mediators, including IL-10 and IDO1 [76,77]. A detailed understanding of these interconnected metabolic and immunological mechanisms is essential for developing therapeutic strategies that reinvigorate DC function and enhance immune responses against HCC. These complex suppressive interactions are summarized in Figure 2, which illustrates how various immune cells and tumor-derived factors inhibit DC function within the HCC microenvironment.

## 4. Dendritic Cell Function Modulation in HCC Therapy

### 4.1. Tumor-Lysate-Pulsed DC Vaccine

The tumor-lysate-pulsed DC vaccine represents an innovative cancer immunotherapy designed to stimulate the immune system to recognize and destroy tumors. This approach utilizes lysates derived from the patients’ tumor cells to activate DCs.

The active components of tumor cell lysates primarily include tumor antigens, tumor-derived DNA and RNA, proteins from lysed tumor cells, cytokines and chemokines, and tumor-specific metabolites. Tumor antigens, defined as proteins or peptides unique to or highly expressed in tumor cells, are key components in activating immune responses and are presented to T cells via DCs to induce specific immune responses [78]. Tumor-derived DNA and RNA can activate the immune system through cellular receptors, thereby enhancing the immune response. Cleaved tumor proteins can also be taken up and processed by DCs to activate T cells. Cytokines and chemokines further enhance the immune response, while tumor-specific metabolites provide additional immune-stimulatory signals [79]. The activated DCs are then reintroduced into the patient to stimulate a robust and targeted antitumor immune response. This personalized approach aims to ensure specificity, reduce immune rejection, and address individual tumor heterogeneity [80].

#### 4.1.1. Autologous Tumor-Lysate-Pulsed DC Vaccines

Autologous tumor-lysate-pulsed DC vaccines contain the full spectrum of antigens from the patient’s own tumor, which ensures antigen specificity and personalization. By delivering antigens through MHC molecules, these vaccines activate both CTLs and helper T cells, initiating a comprehensive immune response [79]. A phase I clinical trial conducted by Iwashita et al. demonstrated the safety and feasibility of a mature DC vaccine pulsed with autologous HCC tumor lysate in patients with unresectable primary HCC. This study showed that tumor-lysate-pulsed DCs were well tolerated and associated with stable disease progression [81]. Additionally, reductions in tumor markers such as alpha-fetoprotein (AFP) were also observed. These results underscore the potential of autologous DC vaccines as a therapeutic strategy, especially for advanced HCC.

#### 4.1.2. Allogeneic Tumor Cell Line Lysate-Pulsed DC Vaccines

Allogeneic tumor cell line lysate-pulsed DC vaccines represent a promising immunotherapeutic strategy that differs from autologous approaches by utilizing lysates derived from standardized tumor cell lines. These cell lines are cultivated under controlled laboratory conditions, ensuring consistency and quality of composition. Since the lysates originate from allogeneic tumor cell lines, they often harbor shared tumor-associated antigens(TAAs), enabling the induction of a broader immune response. This broader response not only enhances immune surveillance by recognizing multiple tumor antigens but also effectively overcomes tumor immune escape mechanisms. Additionally, as the vaccine does not rely on the patients’ own tumor cells, it is suitable for patients who cannot provide sufficient tumor material, thereby expanding clinical applicability [82].

Clinical evidence supports the safety and efficacy of this approach. In a phase II clinical trial conducted by Palmer et al., a mature DC vaccine activated with whole-cell lysates from the HepG2 HCC cell line was evaluated in 35 patients with advanced primary HCC. Patients received several intravenous doses of the vaccine at three-week intervals. The results demonstrated that the DC vaccine was well tolerated in all patients, with no significant adverse effects reported. Some patients exhibited varying degrees of tumor shrinkage or stability, and significant reductions in serum AFP levels were observed in some cases. Longer overall survival was also observed in these patients [83]. Another clinical trial involving patients with advanced HCC similarly confirmed the safety and potential efficacy of this vaccine [84]. These findings collectively highlight the promise of allogeneic tumor cell lines as a viable therapeutic option, especially for HCC patients who cannot be treated with existing therapies, although there remains significant room for improvement in therapeutic efficacy. Future research should focus on optimizing antigen presentation, enhancing DC activation, and identifying strategies to potentiate antitumor immune responses.

#### 4.1.3. Specific-Tumor-Antigen-Pulsed DC Vaccines

Specific-Tumor-Antigen-Pulsed DC Vaccines represent a promising class of immunotherapies designed to activate DCs by loading them with well-defined tumor antigens, including TAAs or tumor-specific antigens (TSAs). Tumor-associated antigens such as AFP and glypican-3 (GPC-3), which are highly expressed in tumor cells, are often targeted in specific immunotherapy for HCC [85]. In addition, neoantigens generated from tumor-specific mutations are highly immunogenic and can be identified by whole-exon sequencing and loaded into DCs to induce anti-tumor T cells. Virus-associated antigens (e.g., HBV core protein HBcAg and HCV nonstructural protein NS3) also exhibit significant immunologic potential in patients with virus-associated HCC [86]. Other potential targets, including HSPs and tumor stem cell markers such as epithelial cell adhesion molecule EpCAM, have also demonstrated roles in promoting antigen delivery and enhancing immune activity, providing multiple options for HCC vaccine research [87,88].

These vaccines aim to induce potent and specific immune responses, enabling precise recognition and eradication of tumor cells. Clinical trials have demonstrated their safety and feasibility. For instance, a phase I/II clinical trial (NCT00022334) conducted by Butterfield et al. evaluated mature DC vaccines pulsed with AFP-derived peptides (AFP137-145, AFP158-166, AFP325-334, AFP542-550) in HLA-A*0201-positive patients with AFP-expressing HCC. Patients received intradermal injections of 1 × 10^6^ to 1 × 10^7^ DCs three times at two-week intervals. The vaccines were well tolerated, and 60% of participants exhibited enhanced IFN-γ-producing AFP-specific CTL responses [89]. These findings underscore the potential of specific-tumor-antigen-pulsed DC vaccines in advancing precision immunotherapy for HCC.

#### 4.1.4. Tumor-Antigen-Pulsed DC Vaccines Combined with Traditional Anticancer Therapies

Tumor-antigen-pulsed DC vaccines have been extensively investigated in clinical trials to evaluate their safety and efficacy when combined with standard anticancer therapies [90]. These trials have demonstrated that DC vaccines are safe and well tolerated as adjuvant treatments [91]. For instance, the autologous tumor lysate-pulsed DC vaccine, when administered alongside hepatic arterial chemoembolization (TACE) and surgical resection, exhibited moderate adverse effects [92]. Additionally, it enhanced peripheral CTL activity and significantly reduced serum AFP levels [93]. Allogeneic tumor cell line lysate-pulsed DC vaccines have also demonstrated efficacy in combination with TACE or radiofrequency ablation (RFA) therapy and were particularly effective in controlling tumor recurrence [94,95]. In addition, DC vaccines loaded with specific antigens such as AFP, MAGE-1, and GPC-3 enhanced IFN-γ-generating CTL responses against tumor antigens in trials combining TACE or surgical resection [96]. These antigen-specific responses were associated with prolonged disease-free survival (DFS) and extended time to tumor progression (TTP) in certain patient subgroups. While significant improvements in overall survival (OS) or recurrence-free survival (RFS) were not consistently observed across all trials, specific patient populations—particularly those with high antigen expression or those who did not receive certain treatments—benefited from the therapeutic efficacy of these combination strategies.

#### 4.1.5. Tumor-Antigen-Pulsed DC Vaccines Together with Other Immunotherapies

Studies have demonstrated that combining DC vaccines pulsed with autologous tumor lysates with immune effector cells and standard anticancer therapies has shown a favorable safety profile and promising therapeutic efficacy in HCC patients [97]. For instance, a phase II clinical trial conducted by Shimizu et al. demonstrated that the combination of CD3-activated T cell (CAT) therapy with DC vaccines significantly prolonged the OS and RFS of patients [98]. A phase I clinical trial conducted by Zhou et al. demonstrated that combining DC vaccines with naïve DCs, cytokine-induced killer (CIK) cells, DC-precision CTLs, and DC-CIK combination therapy was not only safe and feasible but also contributed to a notable reduction in viral DNA levels in certain patients [99]. Furthermore, the combination of mature DC vaccine and CIK with the targeted drug sorafenib has shown superiority in improving patients’ remission rates and prolonging OS [100].

Emerging personalized therapeutic strategies, such as tumor neoantigen-based DC vaccines combined with neoantigen-specific CTLs, have also demonstrated encouraging outcomes. These regimens effectively generate polyclonal neoantigen-specific CTL responses and delay tumor recurrence. However, the results of some clinical trials remain unpublished, and ongoing studies continue to explore these innovative approaches [101].

### 4.2. DCs Pulsed with DNA Constructs

To target HCC, DC vaccines utilizing DNA constructs encoding TAAs represent a promising innovation. These constructs, often delivered through non-pathogenic viral vectors such as recombinant adenoviruses, enable DCs to express specific TAAs.

In a study targeting HCC, the use of recombinant adenoviral vectors to transduce DCs led to the successful development of a potential vaccine based on the cancer/testis antigen HCA661, an antigen specifically expressed in HCC. This antigen was identified by the researchers through the insertion of the genes encoding its full-length cDNA into replication-deficient E1- and E3-deficient adenovirus to transduce immature DCs derived from HLA-A2-positive healthy donors [102]. These transduced DCs displayed high levels of maturation markers, including GM-CSF, IL-4, and monocyte conditioned medium (MCM). Additional poly [I:C] stimulation could further enhance the expression of CD83, HLA-DR, CD80, and CD86, along with loss of endocytosis, and exhibited the characteristics of functionally mature DCs.

Functional analyses revealed that these Ad661-transduced DCs effectively activate CTLs in autologous peripheral blood mononuclear cells (PBMCs), inducing robust HCA661-specific immune responses [102]. In enzyme-linked immunospot (ELISPOT) assays, approximately 400 IFN-γ spots were detected per 40,000 PBMC effector cells, which was significantly higher than that of DCs transduced with the control vector (approximately 100 spots). Quantitative flow cytometry further demonstrated that the HCA661-specific CTLs accounted for 2.3% of PBMC after stimulation by Ad661-transduced DCs, compared to only 0.1–0.4% in controls [102]. In functional validation, these CTLs effectively targeted and lysed the HCA661- and HLA-A2-expressing HCC cell line HepG2, significantly enhancing antitumor responses [103]. These findings quantitatively demonstrated that recombinant adenovirus-mediated HCA661 expression in DCs not only elicited a potent, antigen-specific immune response but also substantially enhanced CTL-mediated killing of HCC cells. This strategy provides a safe, effective, and promising vaccine approach for HCC immunotherapy.

In addition to HCA661, recombinant adenoviruses encoding other target antigens have been used to stimulate immune responses against specific tumor antigens associated with HCC and to enhance CTL responses. For instance, PEG10 (Paternally Expressed Gene 10), a paternally imprinted retrotransposon-derived gene overexpressed in HCC, has been delivered via recombinant adenovirus in DC-based immunotherapy strategies. PEG10-expressing recombinant adenoviruses have been employed to transduce DCs, enabling them to present PEG10-derived antigens and activate antigen-specific cytotoxic T lymphocytes [104].

TEM8 (Tumor Endothelial Marker 8), also known as ANTXR1, is another potential target. It is a surface protein highly expressed on tumor vasculature, particularly in aggressive HCC, and associated with neovascularization. TEM8 recombinant adenovirus targets tumor endothelial markers, disrupting the tumor vasculature and promoting immune system activation [105]. Although TEM8 is not directly expressed by tumor cells, its targeting indirectly contributes to antitumor immunity by impairing tumor angiogenesis and facilitating immune cell infiltration, thereby improving the tumor microenvironment for DC activation and function. Furthermore, CD40L recombinant adenovirus can upregulate co-stimulatory signals on DCs, enhancing their ability to effectively stimulate T cells [106]. These adenoviral-based approaches have demonstrated promising results in activating antigen-specific immune responses and lay the groundwork for further development of immunotherapies for the treatment of HCC.

### 4.3. Fusion of DCs and Tumor Cells

Fusion cell therapy represents an innovative strategy to enhance anti-tumor immune responses by fusing DCs with tumor cells. Gong et al. pioneered the use of polyethylene glycol (PEG) as a fusion agent to achieve the successful fusion of DCs with tumor cells. This technique enables DCs to process and present a diverse array of TAAs through both MHC I and MHC II pathways, thereby activating both CD4^+^ and CD8^+^ T cells [107]. Alternative methods, such as electrofusion, also represent viable approaches in terms of antitumor vaccine efficacy compared to PEG fusion. Advantages of this approach include the ability to present a broad spectrum of known and unknown tumor antigens, thereby increasing the frequency of polyclonal antigen-specific T cells and inducing long-term effective antitumor immunity [108].

Preclinical studies using HCC models have further substantiated the therapeutic potential of this approach. For instance, fusion vaccines combining MHCC97H cells (a human HCC cell line) with DCs expressing α-Gal and endothelin (END) have been shown to significantly enhance antigen-specific CTL responses. These hybrid cells not only present tumor-derived TAAs but also provide additional targets, such as α-Gal and END, to further increase immune specificity and efficacy [109]. Quantitative experiments revealed that MHCC97H/DC fusion cells expressing α-Gal and END secreted significantly higher levels of IL-12p70 and induced robust IFN-γ from CTLs compared to other fusion types. These CTLs exhibited potent killing effects on MHCC97H tumor cells and their associated endothelial cells. In vivo studies demonstrated remarkable therapeutic outcomes, with 73% of treated nude mice surviving beyond 90 days, compared to complete mortality in the control group [109]. This indicates the potential of a fusion cell vaccine to elicit a potent, multifunctional immune response targeting tumor cells and their microenvironment.

### 4.4. Combination Therapy Strategies with Immune Checkpoint Inhibitors

ICIs have become one of the core approaches in HCC immunotherapy by restoring the antitumor activity of effector T cells via blocking the pathway through which tumor cells suppress the immune response. In HCC, the combination of ICIs and DC vaccines represents a synergistic approach to overcome the immunosuppressive tumor microenvironment.

#### 4.4.1. PD-1/PD-L1 Inhibitors

Blocking the PD-1/PD-L1 pathway (e.g., with Nivolumab, Pembrolizumab, or Atezolizumab) prevents T cell exhaustion, enhancing immune responses [110]. Clinical trials have shown that these drugs induce significant and durable tumor remissions in a subset of patients. For instance, in trials such as the CheckMate 040 and KEYNOTE-240, these agents have demonstrated objective remission rates (ORR) of 15% to 20%, including complete remissions (CR) of 1% to 5%, respectively [111]. Combining DC vaccines with PD-1 inhibitors has demonstrated enhanced T cell activation and antigen-specific responses.

#### 4.4.2. CTLA-4 Inhibitors and Their Combination Therapy

CTLA-4 is a critical immune checkpoint molecule that exerts negative immunomodulatory effects, primarily by attenuating initial T cell activation. In HCC therapy, CTLA-4 inhibitors (e.g., Ipilimumab and Tremelimumab) have demonstrated the ability to enhance T cell proliferation and functional activity by blocking CTLA-4-mediated immunosuppressive signaling pathway. Clinical investigations revealed that Tremelimumab exhibited notable anti-tumor activity, with patients achieving a median OS of one year [112].

The therapeutic efficacy of CTLA-4 inhibitors is further augmented when combined with PD-1/PD-L1 inhibitors. For instance, the CheckMate 040 trial demonstrated that the combination of Nivolumab and Ipilimumab achieved an ORR of 32% and a complete remission rate of 8% in second-line treatment [113]. This combination therapy not only significantly extended patients’ median survival but also demonstrated an acceptable safety profile [114]. These findings emphasize the pivotal role of CTLA-4 inhibitors in combination therapy, particularly for patients with advanced-stage HCC, and highlight the potential for integrating these agents into multifaceted treatment strategies to maximize antitumor immune responses.

#### 4.4.3. Other Immune Checkpoint Inhibitors

Beyond PD-1, PD-L1, and CTLA-4, other immune checkpoint molecules such as immunoglobulin and mucin-domain containing-3 (TIM-3) and lymphocyte-activation gene 3 (LAG-3) have been identified as important regulators in immune evasion mechanisms of HCC [115]. ICIs targeting these molecules are currently under investigation and have demonstrated promising preclinical and early clinical results. The therapeutic potential of TIM-3 and LAG-3 inhibitors is further enhanced when used in combination with DC vaccines and PD-1/PD-L1 blockade [116]. These integrated immunotherapy approaches have shown the capacity to synergistically boost antitumor immune responses by targeting multiple immune escape pathways. Preclinical and clinical studies have reported improvements in PFS and OS with these combination strategies, highlighting their potential to address the challenges of tumor heterogeneity and resistance in advanced HCC treatment. These findings support the continued exploration of novel checkpoint targets to expand the repertoire of effective immunotherapeutic options for HCC.

#### 4.4.4. Combination Therapy Strategies

The combined use of ICIs and specific types of DC vaccines has been shown to significantly enhance the efficacy of immunotherapy for HCC. For example, DC vaccines based on autologous tumor lysates improve antigen-presenting capacity by loading patients’ own tumor antigens. When combined with PD-1 inhibitors (e.g., Nivolumab or Pembrolizumab), they effectively deregulate immune checkpoints from effector T cells, resulting in a significant enhancement of effector T cell activation and anti-tumor function [117]. Additionally, the use of DC vaccines loaded with specific HCC antigens (e.g., AFP or GPC-3) in combination with CTLA-4 inhibitors (e.g., Ipilimumab) not only remodeled the TME but also further amplified the immune effects by decreasing the number and function of Tregs and MDSCs [118]. Another study using a DC vaccine loaded with AFP peptides in combination with a PD-1 inhibitor demonstrated enhanced antigen-specific CTL responses and significant tumor load reduction in patients. Collectively, these studies suggest that a combination therapy strategy based on DC vaccines and ICIs can significantly enhance the immunotherapeutic efficacy of HCC through multi-pathway synergism, providing a highly effective and long-lasting therapeutic tool. To facilitate comparison of these approaches, Table 1 provides a consolidated overview of clinical strategies targeting DCs, along with their features and outcomes.

## 5. Conclusions

Modulating DC function offers a transformative approach to improving immunotherapy outcomes in HCC. By restoring DC activation, enhancing antigen presentation, and reprogramming the immunosuppressive TME, strategies aim to overcome the limitations of conventional therapies and leverage the immune system’s capacity to combat cancer.

Clinical studies have demonstrated the potential of various DC-based therapeutic strategies, such as tumor-lysate-pulsed vaccines, specific tumor-antigen-pulsed vaccines, and fusion cell therapies. These approaches have shown promising results, including increased T cell activation, reduced tumor burden, and improved survival rates in patients with advanced HCC. Furthermore, the integration of DC vaccines with ICIs and other systemic therapies demonstrated amplified therapeutic efficacy, enabling robust and durable antitumor immune responses.

Despite these advancements, significant challenges remain in the development and application of DC-based therapies for HCC. The inherent heterogeneity of DC subsets, the complexity of the TME, and variability in patient-specific tumor profiles necessitate further optimization and personalization of these therapeutic approaches. Furthermore, issues related to the scalability and cost-effectiveness of DC-based therapies must be addressed to facilitate their broader clinical implementation.

Future studies should also prioritize the identification of predictive biomarkers to better guide patient stratification and therapeutic decisions. Evidence suggests that markers such as the abundance of XCR1⁺ cDC1s, elevated expression of transcription factors IRF8 and BATF3, and baseline levels of immune checkpoint or co-stimulatory molecules (e.g., PD-L1 and CD86) on DCs may correlate with treatment responsiveness [119]. Furthermore, soluble factors like IL-10 or VEGF, along with circulating microRNAs such as miR-155, are increasingly recognized as indicators of DC dysfunction and could help forecast clinical benefit [120].

Refining DC-targeted approaches, combining them with novel immunotherapeutic modalities, and tailoring treatments to individual patients should also be a focus of future research. Advanced technologies, such as genomic profiling and artificial intelligence, can further aid in identifying optimal therapeutic combinations and predicting patient responses. By effectively addressing these challenges, DC-based immunotherapy holds the potential to revolutionize the treatment landscape for HCC, improving outcomes and offering hope to patients with this challenging malignancy.

## Figures and Tables

**Figure 1 vaccines-13-00496-f001:**
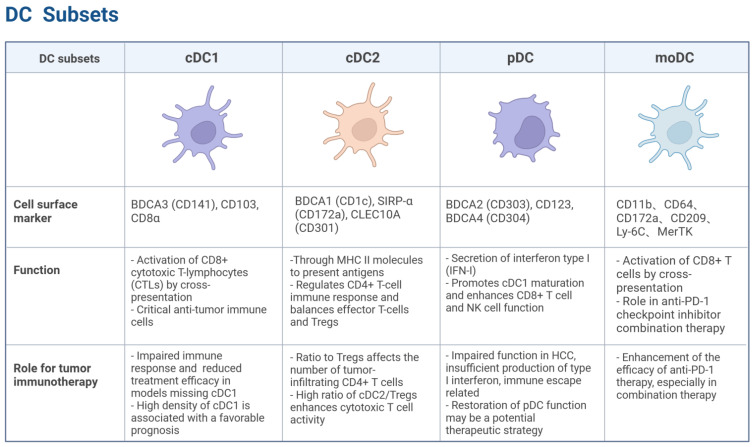
Distinct subsets of DCs. cDC1, characterized by markers such as BDCA3 (CD141), CD103, and CD8α, is crucial for activating CD8^+^ T cells through cross-presentation and plays a significant role in anti-tumor immunity. cDC2, marked by BDCA1 (CD1c) and SIRP-α (CD172a), regulates CD4^+^ T cell responses and balances effector T cells and Tregs. pDCs, identified by BDCA2 (CD303) and CD123, are known for their type I interferon secretion and can enhance anti-tumor immune responses while also promoting immune suppression in the TME. moDCs, marked by CD11b, CD64, and CD209, activate CD8^+^ T cells via cross-presentation and play a role in enhancing anti-PD-1 therapy efficacy. Created with BioRender.com.

**Figure 2 vaccines-13-00496-f002:**
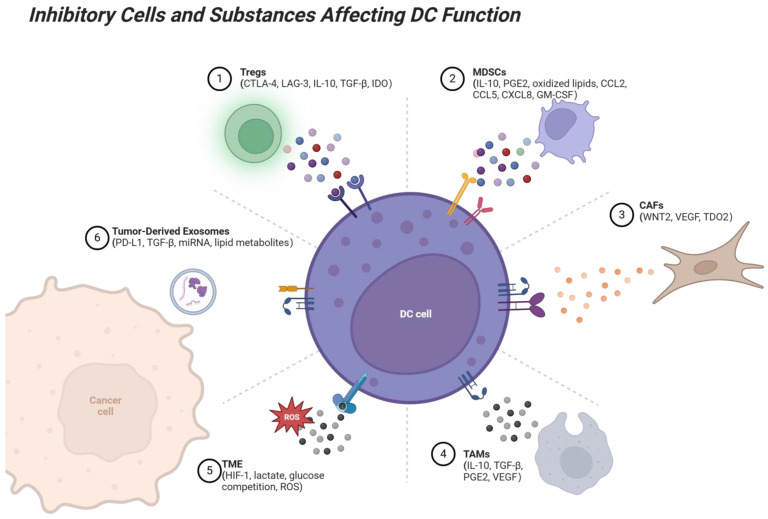
Suppressive interactions between immune cells and DCs in the TME. Several immune cell types, including Tregs, MDSCs, TAMs, CAFs, and tumor-derived exosomes, exert suppressive effects on DCs through various mechanisms. Tregs suppress DC antigen presentation via CTLA-4, LAG-3, and cytokines like IL-10 and TGF-β. MDSCs inhibit DC maturation and cross-presentation via IL-10 and PGE2, while TAMs interfere with DC activation by secreting IL-10, TGF-β, and VEGF. CAFs disrupt DC migration to lymph nodes through VEGF and WNT2 and impair DC antigen presentation through TDO2. Tumor-derived exosomes carry PD-L1, TGF-β, and other suppressive factors, directly inhibiting DC-T cell cooperation. These suppressive interactions limit the ability of DCs to activate T cells, promote immune tolerance, and facilitate tumor immune evasion. Factors within the TME, such as hypoxia, lactate accumulation, and reactive oxygen species (ROS), further contribute to immune suppression. In contrast, enhancing DC function or blocking these suppressive signals represents a promising strategy for overcoming tumor-induced immune suppression. Created with BioRender.com.

**Table 1 vaccines-13-00496-t001:** Modulation of dendritic cells in immunotherapy: methods and clinical outcomes.

Treatment Method	Features	Clinical Trial Identifier	Patient Population	Key Quantitative Outcomes	Key Findings	Reference
Tumor Lysate-Pulsed DC Vaccine (Autologous)	Uses patient-derived tumor lysates to ensure antigen specificity and personalization.	NCT00327496	Unresectable primary HCC	Stable disease progression observed, reductions in tumor markers such as AFP observed.	Well tolerated and feasible.	Iwashita et al., Clinical trial NCT00327496 [81]
Tumor Lysate-Pulsed DC Vaccine (Allogeneic)	Uses standardized tumor cell line lysates for broader antigen coverage; suitable for patients without sufficient tumor samples.	N.A	Advanced primary HCC	Varying degrees of tumor shrinkage or stability observed, significant reductions in serum AFP levels observed in some patients, and longer overall survival observed in patients.	Well tolerated, no significant adverse effects reported	Palmer et al. [83]
Specific Tumor Antigen-Pulsed DC Vaccine	Utilizes specific antigens (e.g., AFP, GPC-3) to induce a highly targeted immune response.	NCT00022334	HLA-A*0201+ AFP-expressing HCC	60% of participants exhibited enhanced interferon-gamma (IFN-γ)-producing AFP-specific cytotoxic T lymphocyte (CTL) responses.	Well tolerated and feasible.	Butterfield et al. [89]
Tumor Antigen-Pulsed DC Vaccine Combined with Cancer Therapies	Combined with traditional anticancer therapies like hepatic arterial chemoembolization (TACE) and surgical resection.	N.A	Patients receiving TACE and surgical resection or TACE/RFA.	Enhanced peripheral cytotoxic T lymphocyte (CTL) activity observed, significantly reduced serum AFP levels observed, and prolonged disease-free survival (DFS) and extended time to tumor progression (TTP) in certain patient subgroups.	Moderate adverse effects, demonstrated efficacy, particularly effective in controlling tumor recurrence (with allogeneic vaccine + TACE/RFA).	Kamigaki et al., Kitahara et al., Lee et al., Maeda et al. [92]
DCs Pulsed with DNA Constructs	Adenoviral vectors delivering tumor antigen genes to DCs to enhance DC antigen expression.	N.A	In vitro (healthy donors, HepG2 cell line) and in vivo (murine models).	HCA661-specific CTLs accounted for 2.3% of PBMC after stimulation by Ad661-transduced DCs in vitro, with potent killing effects on HepG2 cells in vitro, and 73% of treated nude mice surviving beyond 90 days in vivo (with fusion cells).	Successfully developed potential vaccines, elicited potent antigen-specific immune responses, and significantly enhanced antitumor responses.	Chan et al., Peng et al., Yang et al., Guo et al., He et al. [102,104,105,106,109]
DC-Tumor Cell Fusion Vaccine	Fusion of DCs and tumor cells (e.g., via PEG or electrofusion) to present a broad range of antigens.	N.A	In vitro (MHCC97H cells) and in vivo (nude mice).	Secreted significantly higher levels of IL-12p70 and induced robust IFN-γ from CTLs in vitro, exhibited potent killing effects on tumor cells and associated endothelial cells in vitro, and 73% of treated nude mice survived beyond 90 days in vivo.	Elicits potent, multifunctional immune response targeting tumor cells and their microenvironment.	Gong et al., He et al. [107,109]
DC Vaccine Combined with PD-1/PD-L1 Inhibitors	Combination with checkpoint inhibitors to prevent T cell exhaustion and enhance anti-tumor response.	CheckMate 040	Advanced HCC	Objective remission rates (ORR) of 15% to 20% reported for PD-1 inhibitors alone. Combination of Nivolumab + Ipilimumab achieved ORR of 32% and CR of 8% in second-line treatment.	Enhanced T cell activation and antigen-specific responses.	El-Khoueiry et al., Yau et al. [111,113]
DC Vaccine Combined with CTLA-4 Inhibitors	Combined with PD-1 inhibitors to enhance T cell activity and reduce immunosuppressive signaling.	CheckMate 040	Advanced HCC previously treated with Sorafenib.	Combination of Nivolumab and Ipilimumab achieved an ORR of 32% and a complete remission rate of 8% in second-line treatment, significantly extended patients’ median survival.	Remodeled the tumor immune microenvironment, decreased the number and function of Tregs and MDSCs, acceptable safety profile.	Yau et al. [113]
Other Checkpoint Inhibitor Combinations	TIM-3 and LAG-3 inhibitors combined with DC vaccines and/or PD-1/PD-L1 blockade.	N.A	Under investigation.	Preclinical and early clinical studies reported improvements in PFS and OS.	Synergistically boost anti-tumor immune responses by targeting multiple immune escape pathways.	Cheng et al. [116]

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
