# Peer review of "Harnessing Dendritic Cell Function in Hepatocellular Carcinoma: Advances in Immunotherapy and Therapeutic Strategies"

_vaccines, 2025, doi:10.3390/vaccines13050496_

Round 1
Reviewer 1 Report
Comments and Suggestions for Authors
This manuscript reviews the role and mechanism of dendritic cells (DCs) in initiating and modulating immune responses against hepatocellular carcinoma (HCC), and the synergistic effects of DC vaccines combined with traditional anticancer therapies, other immunotherapies, immune checkpoint inhibitors, etc. In addition, the challenges of applying DC-based therapies to clinical practice are also reviewed.
This manuscript is well structured and clearly layered. However, certain sections require deeper mechanistic analysis, updated clinical data, and enhanced clarity. My recommendation is that this paper cannot be published in its current form for several reasons. Here are my comments:
- The interaction between DCs and TME metabolic regulation is only briefly mentioned. How does metabolic reprogramming of DCs influence antitumor immunity? Please elaborate it.
- The clinical data presented in the manuscript are somewhat fragmented, with varying levels of detail. A more consistent format for presenting clinical trial results would enhance readability and comparability.
- Early clinical trials ( Refs. 87, 89) should be updated with recent Phase III data.
- Figure 1 should be Table 1.
- Figure 2 and original Table 1 are not mentioned in the text.
- miRNA, lipid metabolites and ROS shown in Figure 2 are not described in the text.
- Page 11, what are PEG10 and TEM8 respectively? What do they have to do with DC?
- Supplement Virus-associated HCC and DC dysfunction.
- Alternate use of "immune checkpoint inhibitors" and "ICIs"—standardize abbreviations upon first mention.
- Some references lack page numbers.
11.Some wordy sentence(e.g., Section 3.1 on Treg mechanisms)lack logical transitions. Simplify and add subheadings for readability.
- Please refine language for clarity and consistency.
- Introduction: The introduction is well-written and provides a clear rationale for the focus on DCs in HCC immunotherapy. It would be better if it could briefly mention some of the most recent breakthroughs or ongoing challenges in HCC treatment to set the stage for the review.
- Biological functions of dendritic cells and their role in cancer immunity: The section on DC subsets is comprehensive. However, the discussion on cDC2 could benefit from a clearer explanation of the dual role they play in balancing effector T cells and Tregs. Perhaps a diagram or a more detailed example would help clarify this concept for readers.
- DC Function Modulation in HCC Therapy:The discussion on various DC-based therapeutic strategies is thorough. However, the section on DC vaccines combined with other immunotherapies could benefit from a more detailed analysis of the mechanisms underlying the observed synergistic effects. The clinical trial data presented in this section are somewhat inconsistent. A standardized format for presenting trial outcomes would improve clarity and comparability.
16. Conclusions:The conclusions are well-written and provide a good summary of the key points discussed in the manuscript. However, the discussion on future directions should include a brief mention of potential biomarkers or predictive factors that could help identify patients most likely to benefit from DC-based therapies.
Comments on the Quality of English LanguageThe English could be improved to more clearly express the research.
Author Response
This manuscript reviews the role and mechanism of dendritic cells (DCs) in initiating and modulating immune responses against hepatocellular carcinoma (HCC), and the synergistic effects of DC vaccines combined with traditional anticancer therapies, other immunotherapies, immune checkpoint inhibitors, etc. In addition, the challenges of applying DC-based therapies to clinical practice are also reviewed.
This manuscript is well structured and clearly layered. However, certain sections require deeper mechanistic analysis, updated clinical data, and enhanced clarity. My recommendation is that this paper cannot be published in its current form for several reasons. Here are my comments:
1.The interaction between DCs and TME metabolic regulation is only briefly mentioned. How does metabolic reprogramming of DCs influence antitumor immunity? Please elaborate it.
Response: We thank the reviewer for these suggestions. We have discussed the metabolic reprogramming of DC in the TME in the new version of the manuscript.
Page 9:
Dendritic cells (DCs), pivotal orchestrators of anti-tumor immunity, are also pro-foundly affected by these metabolic constraints. Within the nutrient-depleted and hy-poxic TME, DCs undergo metabolic reprogramming that skews them toward a tolero-genic phenotype. Elevated lipid uptake, mediated by receptors like CD204, disrupts MHC class I-mediated antigen presentation and impairs T cell priming[74]. Further-more, a shift of DCs from glycolysis to fatty acid oxidation (FAO)—driven by PPARs and β-catenin signaling—facilitates Treg induction and inhibits cytotoxic T cell re-sponses[75]. Metabolites such as lactate and adenosine reinforce this suppressive state by downregulating pro-inflammatory cytokines like IL-12 and upregulating immuno-regulatory mediators including IL-10 and IDO1[76, 77]. A detailed understanding of these interconnected metabolic and immunological mechanisms is essential for devel-oping therapeutic strategies that reinvigorate DC function and enhance immune re-sponses against HCC. These complex suppressive interactions are summarized in Fig-ure 1, which illustrates how various immune cells and tumor-derived factors inhibit DC function within the HCC microenvironment.
2.The clinical data presented in the manuscript are somewhat fragmented, with varying levels of detail. A more consistent format for presenting clinical trial results would enhance readability and comparability.
Response: We thank the reviewer for this insightful comment. To address concerns regarding consistency, we have revised the Table 2 by providing additional information and would like to highlight that Table 2 was designed to provide a standardized summary of all clinical trials discussed, including the treatment methods, specific features, clinical outcomes with quantitative data, and literature citations. This table aims to facilitate direct comparison and enhance overall readability.
Regarding the varying levels of detail in the main text, this reflects the heterogeneity in data availability and reporting standards across different studies. In some trials, comprehensive outcome metrics (e.g., CTL response rates, AFP level changes, survival data) are provided, while in others, only limited information such as safety or feasibility is available. Our intention was to report the most relevant and reliable data for each study without overinterpreting findings or introducing bias due to incomplete datasets. This differential treatment ensures scientific rigor while preserving the integrity and granularity of each study’s contribution to the field. We hope this rationale and the structured format of Table 2 address the reviewer’s concern.
3.Early clinical trials ( Refs. 87, 89) should be updated with recent Phase III data.
Response: We thank the reviewer for this valuable suggestion. In response, we conducted a comprehensive literature search to identify recent Phase III clinical trial data specifically related to DC vaccines in HCC. However, at the time of our review, we were unable to locate any published Phase III trials that directly build upon or update the findings presented in Refs. 87 and 89. The majority of available studies remain in early-phase development (Phase I or II), and although some trials are ongoing or have been registered, results have not yet been made publicly available or peer-reviewed.
4.Figure 1 should be Table 1.
Response: Thank you for the correction. We have updated the label from “Figure 1” to “Table 1” as suggested.
5.Figure 2 and original Table 1 are not mentioned in the text.
Response: We thank the reviewer for highlighting this oversight. We have now revised the manuscript to ensure that both Figure 2 and the original Table1 are appropriately cited and discussed within the main text.
6.miRNA, lipid metabolites and ROS shown in Figure 2 are not described in the text.
Response: We thank the reviewer for this important observation. In response, we have revised the manuscript to incorporate a brief explanation of the roles of miRNAs, lipid metabolites, and reactive oxygen species (ROS) in modulating dendritic cell function within the tumor microenvironment.
Page 8-9:
Tumor-Derived Exosomes
One significant contributor to the immunosuppressive TME is the presence of tu-mor-derived exosomes. These small vesicles secreted by tumor cells carry a variety of immunosuppressive factors that can alter DC function and hinder their ability to effec-tively activate T cells. Tumor-derived exosomes deliver molecules such as PD-L1, TGF-β, miRNAs, and lipid metabolites that significantly impact DC behavior[66]. For example, PD-L1 on tumor-derived exosomes binds to PD-1 on DCs, leading to a suppression of DC maturation and function, which ultimately dampens T cell activation. Similarly, TGF-β delivered from exosomes promotes the polarization of DCs into an immunosup-pressive phenotype, reducing their capacity to activate effector T cells. Furthermore, miRNAs such as miR-21 and miR-155, carried by exosomes, have been shown to regu-late the differentiation and function of DCs, favoring the expansion of immunosup-pressive cell populations like MDSCs and limiting T cell responses[67]. Additionally, li-pid metabolites within tumor-derived exosomes, such as PGE2, can influence DC mat-uration and function, contributing to the suppression of antitumor immunity[68]. Col-lectively, these tumor-derived exosomes effectively reprogram DCs in the TME, im-pairing their antigen-presenting ability and promoting an environment conducive to tumor immune evasion. By interfering with DC functions, exosomes not only prevent effective immune responses but also contribute to the overall immunosuppressive landscape in HCC[69].
7.Page 11, what are PEG10 and TEM8 respectively? What do they have to do with DC?
Response: We thank the reviewer for raising up this point. We have revised our manuscript and described PEG10 and TEM8 in detail.
Page 13-14:
In addition to HCA661, recombinant adenoviruses encoding other target antigens have been used to stimulate immune responses against specific tumor antigens associ-ated with HCC and to enhance CTL responses. For instance, PEG10 (Paternally Ex-pressed Gene 10), a paternally imprinted retrotransposon-derived gene overexpressed in HCC, has been delivered via recombinant adenovirus in DC-based immunotherapy strategies. PEG10-expressing recombinant adenoviruses have been employed to trans-duce DCs, enabling them to present PEG10-derived antigens and activate anti-gen-specific cytotoxic T lymphocytes.[104].
TEM8 (Tumor Endothelial Marker 8), also known as ANTXR1, is another potential target. It is a surface protein highly expressed on tumor vasculature, particularly in ag-gressive HCC, and associated with neovascularization. TEM8 recombinant adenovirus target tumor endothelial markers, disrupting the tumor vasculature and promoting immune system activation[105]. Although TEM8 is not directly expressed by tumor cells, its targeting indirectly contributes to antitumor immunity by impairing tumor angio-genesis and facilitating immune cell infiltration, thereby improving the tumor micro-environment for DC activation and function. Furthermore, CD40L recombinant adeno-virus can upregulate co-stimulatory signals on DCs, enhancing their ability to effectively stimulate T cells[106]. These adenoviral-based approaches have demonstrated promising results in activating antigen-specific immune responses and lay the groundwork for further development of immunotherapies for the treatment of HCC.
8.Supplement Virus-associated HCC and DC dysfunction.
Response: We thank the reviewer for this suggestion. We have discussed virus-associated HCC and DC dysfunction in Page 8.
3.2 DC dysfunction in virus-associated HCC
Chronic hepatitis B virus (HBV) infection, a major etiological factor for HCC, is characterized by profound immune dysregulation, including DC dysfunction. Accumu-lating evidence indicates that both moDCs and pDCs from HBV-infected individuals exhibit impaired phenotypic and functional maturation. These defects include reduced expression of costimulatory molecules (CD80, CD86, HLA-DR) and diminished secretion of IL-12, which collectively compromise T-cell priming and cytotoxic effector responses[57]. Concurrently, HBV-associated DCs produce higher levels of immunosuppressive mediators such as nitric oxide and IL-10, further contributing to immune tolerance and viral persistence[58].
HBV surface antigen (HBsAg) has been shown to interact directly with DCs, lead-ing to aberrant activation. While it can promote partial maturation through sCD14 and TLR4-dependent pathways, the resulting DCs exhibit a tolerogenic phenotype with im-paired IL-12 production and weakened T cell stimulation[59]. Moreover, transcriptomic profiling has revealed a downregulation of IL6ST (gp130), a core component of IL-6 family cytokine signaling, in DCs from chronic HBV patients. Experimental restoration of IL6ST in vitro partially reversed the functional defects of these DCs, suggesting a mechanistic link between HBV infection and disrupted cytokine receptor signaling[60]. These alterations in DC function not only facilitate immune evasion by HBV but also create an immunologically permissive environment that may accelerate hepatocarcinogenesis.
9.Alternate use of "immune checkpoint inhibitors" and "ICIs"—standardize abbreviations upon first mention.
Response: Thank you for your valuable suggestion. We have revised the manuscript to ensure that "immune checkpoint inhibitors (ICIs)" is introduced at the first mention, and the abbreviation "ICIs" is consistently used thereafter throughout the text.
10.Some references lack page numbers.
Response: "Thank you for pointing out the need for completeness in the reference list. We have carefully reviewed all references. For journal articles that use traditional pagination, we have ensured page numbers are included. We noted that several of the cited articles are from online-only journals or formats that utilize article numbers (or eLocators) rather than sequential page numbers. For these specific references, traditional page numbers are not applicable. However, we have verified and included the correct article identifiers and DOIs for these entries to ensure accurate identification and retrieval. We have ensured the formatting is consistent throughout the reference list according to the journal's guidelines.
11.Some wordy sentence(e.g., Section 3.1 on Treg mechanisms)lack logical transitions. Simplify and add subheadings for readability.
Response: Thank you for your thoughtful feedback. We have carefully revised Section 3.1 by simplifying wordy sentences, improving logical transitions between ideas, and adding subheadings to enhance clarity and readability.
12.Please refine language for clarity and consistency.
Response: Thank you for your suggestion. We have thoroughly revised the manuscript to enhance clarity, improve consistency, and ensure more precise use of terminology throughout the text.
13.Introduction: The introduction is well-written and provides a clear rationale for the focus on DCs in HCC immunotherapy. It would be better if it could briefly mention some of the most recent breakthroughs or ongoing challenges in HCC treatment to set the stage for the review.
Response: We thank the reviewer for this insightful comment, we have described this in our revised manuscript.
Page 1-2:
Recent breakthroughs, particularly the establishment of immune checkpoint inhibitors (ICIs) combinations such as atezolizumab plus bevacizumab, durvalumab plus tremelimumab as standard first-line treatments, have significantly improved survival outcomes for patients with advanced HCC[1]. Furthermore, novel strategies combining these systemic agents with locoregional therapies, such as transcatheter arterial chemoembolization (TACE) are emerging as effective approaches for intermediate-stage disease[2]. Despite these advances, significant challenges persist: many patients do not respond to current ICI-based regimens, treatment resistance frequently develops, and overcoming the profoundly TME remains a critical barrier to efficacy. Addressing the limitations of current therapies motivates the exploration of alternative approaches; harnessing the unique capabilities of dendritic cells (DCs) represents a promising immunotherapeutic direction.
14.Biological functions of dendritic cells and their role in cancer immunity: The section on DC subsets is comprehensive. However, the discussion on cDC2 could benefit from a clearer explanation of the dual role they play in balancing effector T cells and Tregs. Perhaps a diagram or a more detailed example would help clarify this concept for readers.
Response: We thank the reviewer for this insight thought. We have added this into our revised manuscript.
Page 2-4:
cDC2 The cDC2 subset of DCs plays a crucial yet complex role in shaping the an-ti-tumor immune response. While traditionally recognized for their ability to prime CD4+ T helper cells that support cytotoxic T cell responses against cancer, recent re-search highlights a dual capacity of cDC2s: they can also contribute to the induction and maintenance of regulatory T cells (Tregs), thereby potentially suppressing anti-tumor immunity. Understanding this delicate balance is critical for developing effective cancer immunotherapies.
cDC2s are highly efficient at capturing and presenting exogenous antigens, including those derived from tumors, primarily to CD4+ T cells via MHC class II molecules[11]. Upon activation in a pro-inflammatory environment, typically characterized by the presence of danger signals and specific cytokines, cDC2s can effectively prime naive CD4+ T cells to differentiate into various effector T helper subsets, such as Th1 and Th17 cells[12]. Th1 cells are particularly important in anti-tumor immunity as they produce cytokines like interferon-gamma (IFN-γ) that enhance the activity of cytotoxic CD8+ T cells, the primary killers of cancer cells. Th17 cells can also contribute to anti-tumor responses through the recruitment of other immune cells. This pathway rep-resents the immunogenic facet of cDC2 function, promoting a robust anti-cancer immune response. Studies have shown that cDC2s are vital for priming anti-tumor CD4+ T cells in various tumor models[3].
However, the TME is often enriched with immunosuppressive factors, including certain cytokines (e.g., IL-10, TGF-β) and metabolic conditions, which can significantly influence DC function[13, 14]. In this altered state, cDC2s can adopt a more tolerogenic phenotype. Instead of driving effector T cell differentiation, they can promote the in-duction of Tregs. In the context of cancer, an increased presence and activity of Tregs within the TME can suppress the function of effector T cells, including tumor-specific CTLs, thereby allowing the tumor to evade immune surveillance and progress. cDC2s can contribute to Treg induction through various mechanisms, including the presentation of specific antigens in a non-inflammatory context, the production of immunosuppressive cytokines, and the expression of co-inhibitory molecules. The plasticity of cDC2s in response to tumor-released environmental cues has been observed, leading to the acquisition of immunosuppressive phenotypes[15].
This dual capacity of cDC2s underscores their critical position at the intersection of immunity and tolerance in the context of cancer. Their ultimate effect on the anti-tumor response depends heavily on the signals they receive from their microenvironment. A pro-inflammatory milieu favors their role in promoting effector T cells, while an immunosuppressive TME can skew them towards inducing Tregs.
In melanoma, a tumor type often characterized by significant immune infiltration and referred to as an "inflamed" or "hot" tumor. In this context, the TME is rich in pro-inflammatory cytokines like IFN-γ and TNF-α, secreted by infiltrating immune cells such as NK cells and activated T cells. Intratumoral cDC2s within this environment efficiently capture tumor-derived antigens. The robust inflammatory signals drive their maturation, leading to the upregulation of co-stimulatory molecules and MHC class II. These activated cDC2s then migrate to the regional lymph nodes to induce the differentiation of naive CD4+ T cells into effector Th1 cells. These tumor-specific Th1 cells, armed with the ability to produce IFN-γ and IL-2, subsequently traffic back to the tumor, where they significantly enhance the activity of tumor-infiltrating cytotoxic CD8+ T cells, contributing to tumor cell lysis and control[16].
Conversely, within immunosuppressive TMEs, such as those frequently observed in cancers like pancreatic carcinoma, the function of cDC2s can be significantly skewed towards promoting tolerance. These environments are characterized by high concentrations of immunosuppressive factors, including TGF-β, IL-10, and vascular endothelial growth factor (VEGF), often accompanied by a dense stromal barrier and metabolic constraints. In such settings, cDC2s exhibit impaired maturation, reduce expression of co-stimulatory molecules, and increase expression of inhibitory ligands such as programmed death ligand 1 (PD-L1)[11]. The cytokine profile of the immunosuppressive TME favors the induction and expansion of Tregs.
This inherent plasticity and dual capacity of cDC2s highlight their central position in the complex interplay between the tumor and the immune system. Consequently, a comprehensive understanding of the specific molecular and cellular cues within the TME that dictate cDC2 polarization towards either immunogenic or tolerogenic functions is therefore paramount. Manipulating this balance, either by directly targeting cDC2 function to enhance effector priming or by remodeling the immunosuppressive TME to favor cDC2 maturation and immunogenicity, represents a promising avenue for improving the efficacy of current and future cancer immunotherapies.
15.DC Function Modulation in HCC Therapy: The discussion on various DC-based therapeutic strategies is thorough. However, the section on DC vaccines combined with other immunotherapies could benefit from a more detailed analysis of the mechanisms underlying the observed synergistic effects. The clinical trial data presented in this section are somewhat inconsistent. A standardized format for presenting trial outcomes would improve clarity and comparability.
REPONSE: We thank the reviewer for providing this insightful suggestion. We have revised the Table to provide a more detailed information.
- Conclusions: The conclusions are well-written and provide a good summary of the key points discussed in the manuscript. However, the discussion on future directions should include a brief mention of potential biomarkers or predictive factors that could help identify patients most likely to benefit from DC-based therapies.
Response: we thank the reviewer for this point. We have revised our conclusion.
Reviewer 2 Report
Comments and Suggestions for Authors
Ying’s manuscript aimed to dissect the role of dendritic cells (DCs) in hepatocellular carcinoma (HCC) and discuss advancements in immunotherapy and therapeutic strategies. This would provide scientists and clinicians with techniques for DC manipulation that could potentially assist them in utilizing DCs to combat HCC. However, the review requires significant revision to meet the authors' initial aim.
Current knowledge of DC immunobiology must be concise and integrated into the context of HCC. The manuscript's purpose is not to address general DC biology or the role of DCs in cancers. If the authors wish to support their arguments, they could use this knowledge to substantiate their claims, ideas, proposals, and arguments. Therefore, Section 2 must be completely revised based on this concept.
The same reasoning applies to Section 3, Impaired DC Functions in the Tumor Microenvironment (TME). This section must be revised to examine impaired DC functions, specifically within the HCC microenvironment.
- The conclusion is vague and lacks controversies, gaps, and definitive proposals to advance research in the field. Like the other sections of the manuscript, the conclusion lacks critical analysis and intellectual input.
Author Response
Ying’s manuscript aimed to dissect the role of dendritic cells (DCs) in hepatocellular carcinoma (HCC) and discuss advancements in immunotherapy and therapeutic strategies. This would provide scientists and clinicians with techniques for DC manipulation that could potentially assist them in utilizing DCs to combat HCC. However, the review requires significant revision to meet the authors' initial aim.
1.Current knowledge of DC immunobiology must be concise and integrated into the context of HCC. The manuscript's purpose is not to address general DC biology or the role of DCs in cancers. If the authors wish to support their arguments, they could use this knowledge to substantiate their claims, ideas, proposals, and arguments. Therefore, Section 2 must be completely revised based on this concept.
Response: Thank you for your careful review of our manuscript and for your specific comments on Section 2. We appreciate your perspective regarding the need for the discussion on DC immunobiology to be concise and tightly integrated into the context of HCC, focusing on how this knowledge directly supports the arguments presented in the review.
We have carefully considered your recommendation for a complete revision of Section 2. While we understand your point about maintaining a sharp focus on HCC, we believe that providing a foundational overview of the different DC subsets and their core functions in initiating adaptive immunity is essential for readers from diverse backgrounds to fully appreciate the subsequent detailed discussion on how DC function is specifically impaired in the complex HCC microenvironment and how this impairment informs the rationale for DC-based therapeutic strategies.
Our intention in Section 2 was to lay the necessary groundwork by briefly introducing the key players (DC subsets) and fundamental processes of DC biology before delving into the specific challenges and opportunities in HCC. We feel that omitting or significantly condensing this foundational information might make it difficult for some readers to fully grasp the intricacies of DC dysfunction in HCC and the mechanisms by which DC-based therapies aim to restore or enhance anti-tumor immunity in this specific cancer setting.
Therefore, while we will not be able to completely revise Section 2 as requested, we propose to make targeted revisions to enhance the integration with the HCC context. We will add clearer transitional phrases and sentences throughout Section 2 to explicitly link the general aspects of DC biology to their relevance in HCC. We will also review the section to identify any areas where the language can be slightly adjusted to emphasize the connection to cancer immunity and, where appropriate, specifically to HCC, without losing the necessary foundational detail.
We believe that these targeted revisions will help to address your concern about the integration with the HCC context while preserving the essential introductory information that supports the main themes of the review regarding impaired DC function and therapeutic modulation in HCC.
Thank you again for your understanding.
2.The same reasoning applies to Section 3, Impaired DC Functions in the Tumor Microenvironment (TME). This section must be revised to examine impaired DC functions, specifically within the HCC microenvironment.
Response: We thank the reviewer for this insightful suggestion, we have carefully revised the Section 3.
3.The conclusion is vague and lacks controversies, gaps, and definitive proposals to advance research in the field. Like the other sections of the manuscript, the conclusion lacks critical analysis and intellectual input.
Response: We thank the reviewer for pointing out this. We have carefully revised the Conclusion section.
Round 2
Reviewer 2 Report
Comments and Suggestions for Authors
The authors applied my suggestions and addressed my concerns satisfactorily.